# Updated Review and Clinical Recommendations for the Diagnosis and Treatment of Patients with Retroperitoneal Sarcoma by the Spanish Sarcoma Research Group (GEIS)

**DOI:** 10.3390/cancers15123194

**Published:** 2023-06-15

**Authors:** Rosa Álvarez Álvarez, Aránzazu Manzano, Carolina Agra Pujol, Vicente Artigas Raventós, Raquel Correa, Josefina Cruz Jurado, Juan Angel Fernandez, Xavier Garcia del Muro, Jose Antonio Gonzalez, Nadia Hindi, Pablo Lozano Lominchar, Javier Martínez-Trufero, Ramiro Méndez, Mercedes Muñoz, Cristobal Muñoz Casares, Francisco Orbis Castellanos, Ruth Orellana Fernandez, Miguel Paniagua González, Andres Redondo, Claudia Valverde Morales, Jose Manuel Asencio

**Affiliations:** 1Medical Oncology Department, Instituto de Investigacion Sanitaria Gregorio Marañon, Hospital Universitario Gregorio Marañon, 28007 Madrid, Spain; 2Medical Oncology Department, Hospital Universitario Clínico San Carlos, 28040 Madrid, Spain; aranzazu.manzano@salud.madrid.org; 3Pathology Department, Hospital Universitario Gregorio Marañon, Complutense University, 28007 Madrid, Spain; caroagra@gmail.com; 4Surgery Department, Hospital Universitario Sant Pau, Universidad Autonoma de Barcelona, 08035 Barcelona, Spain; vartigas18@gmail.com (V.A.R.); jgonzalezl@santpau.cat (J.A.G.); 5Radiation Oncology Department, Hospital Virgen de la Victoria, 29010 Malaga, Spain; racoge@hotmail.com; 6Medical Oncology Department, Hospital Universitario Canarias, 38320 Santa Cruz de Tenerife, Spain; jcruzjurado@gmail.com; 7Sarcoma Multidisciplinary Unit, Surgery Department, Hospital Virgen de la Arrixaca, 30120 Murcia, Spain; jaferher@outlook.com; 8Sarcoma Multidisciplinary Unit, Medical Oncology Department, Idibell, Instituto Catalan Oncología Hospitalet, 08908 Barcelona, Spain; garciadelmuro@iconcologia.net; 9Medical Oncology Department, Fundacion Jimenez Diaz University Hospital, 28040 Madrid, Spain; nhindi@atbsarc.org; 10Medical Oncology Department, Hospital General de Villalba, 28400 Madrid, Spain; 11Health Research Institute-Fundación Jiménez Díaz (IIS-FJD), Universidad Autónoma de Madrid (UAM), 28040 Madrid, Spain; 12Surgery Department, Hospital Universitario Gregorio Marañon, Complutense University, 28040 Madrid, Spain; lozanon57@hotmail.com (P.L.L.); jmasencio@gmail.com (J.M.A.); 13Medical Oncology Department, Hospital Universitario Miguel Servet, 50009 Zaragoza, Spain; jmtrufero@seom.org; 14Radiology Department, Hospital Universitario Clínico San Carlos, 28040 Madrid, Spain; ramiro.mendez@salud.madrid.org; 15Radiation Oncology Department, Hospital Universitario Gregorio Marañon, Complutense University, 28007 Madrid, Spain; memepm@hotmail.com; 16Surgery Department, Hospital San Juan de Dios, 14012 Cordoba, Spain; fcocris@gmail.com; 17Sarcoma Multidisciplinary Unit, Surgery Department, Hospital Universitario y Politécnico La Fe, 46026 Valencia, Spain; orbisp@yahoo.es; 18Pathology Department, Hospital Universitario Santa Creu i Sant Pau, 08025 Barcelona, Spain; rorellana@santpau.cat; 19Radiology Department, Hospital Universitario Gregorio Marañon, Complutense University, 28007 Madrid, Spain; m.paniagua.gonzalez@gmail.com; 20Medical Oncology Department, Hospital Universitario La Paz—IdiPAZ, 28046 Madrid, Spain; andres.redondos@uam.es; 21Medical Oncology Department, Vall D’Hebron University Hospital, 08035 Barcelona, Spain; cvalverde@vhio.net

**Keywords:** retroperitoneal sarcoma, soft tissue sarcoma, retroperitoneum, multidisciplinary board, reference centers

## Abstract

**Simple Summary:**

The treatment of retroperitoneal sarcomas poses significant challenges due to their infrequency, clinical and histologic heterogeneity, and unique anatomical location. We provide an updated review on the management of retroperitoneal sarcoma and issue clear and concise recommendations for the treatment of the main clinical situations encountered in this disease.

**Abstract:**

Soft tissue sarcomas (STS) are an uncommon and biologically heterogeneous group of tumors arising from mesenchymal cells. The incidence is estimated at five cases per 100,000 people per year. Retroperitoneal sarcomas (RPS) account for 10–15% of all STS, and their management depends on their anatomical characteristics and histotype. Due to their very low incidence, it is recommended that RPS be treated in reference centers and evaluated by an experienced multidisciplinary team (MDT). In Spain, the Spanish Group for Research in Sarcomas (GEIS) brings together experts from various specialties to promote research on sarcomas and improve treatment results. This paper summarizes the GEIS recommendations for the diagnosis, treatment, and follow-up of patients with RPS.

## 1. Introduction

Soft tissue sarcomas (STS) are an uncommon and heterogeneous group of tumors of mesenchymal cell origin, with an estimated incidence of five cases per 100,000 per year in Europe [1,2]. Approximately 10–15% of all STS are retroperitoneal sarcomas (RPS). RPS usually presents at an advanced stage with nonspecific symptoms, such as increased abdominal perimeter, abdominal pain, and a change in bowel habits. Although STS comprise more than 100 histopathologic subtypes, in the retroperitoneum, the most frequent subtypes are, in order of frequency, well-differentiated liposarcoma (WDLPS)/dedifferentiated liposarcoma (DDLPS), leiomyosarcoma (LMS), solitary fibrous tumor (SFT) and malignant peripheral nerve sheath tumor (MPNST) [3]. Each of these entities has its own distinct biological behavior in terms of risks of local (LR) or distant recurrence (DR) and overall survival (OS) [4]. Recent advances in the understanding of the biological variability of RPS have led to more personalized histology-based management that includes surgical and non-surgical treatments such as radiotherapy and chemotherapy.

RPS requires a multidisciplinary and complex therapeutic approach and should preferably be treated in specialized centers with a team of radiologists, pathologists, surgeons, and medical and radiation oncologists with expertise in the treatment of this disease. For this reason, there has been increasing interest in centralizing the management of these patients in national reference centers. In addition, international cooperation has led to the creation of collaborative groups, such as the Trans-Atlantic Australasian Retroperitoneal Sarcoma Working Group (TARPSWG), to improve knowledge of this disease and advance its treatment.

GEIS has elaborated its first multidisciplinary clinical practice guidelines for the disease to provide clear and concise recommendations on the main clinical situations encountered in RPS. We consider them useful in an integrated multidisciplinary approach to management, and they contribute to improving the care of patients with this diagnosis.

## 2. Methodology

This guideline was developed by a multidisciplinary group of specialists from different fields involved in the diagnosis and treatment of RPS. We systematically searched data from PUBMED, EMBASE, CENTRAL, OpenGrey, ClinicalTrials.gov, and ProQuest. In each section, we performed different searches, in order to address different questions, prioritizing data with the best evidence level. After that, in a consensus meeting, each section was presented by an expert to the whole group for discussion. The three coordinating authors (RA, JMA, and AM) were responsible for compiling and homogenizing the various sections. All authors reviewed and approved the final version of the document. The panel adopted the Infectious Disease Society of America levels of evidence (I to V) and grades of recommendation (A to C) [5] (Table 1).

## 3. Warning Signs and Indications for Referral to Specialist Sarcoma Centers—What Can the Reference Center Offer?

RPS is frequently diagnosed after incidental findings on CT scans performed for other reasons. If symptoms are present, they tend to be nonspecific, such as abdominal pain or back pain, or compression of other organs (bowel obstruction, urinary or gynecological symptoms), and they are usually associated with advanced disease. In cases of bulky disease, an abdominal mass can be palpated.

As RPS are rare tumors and surgery is the cornerstone of treatment, it is not surprising that patients undergoing resection of primary RPS within a specialist sarcoma center not only have less early postoperative morbidity and a lower risk of postoperative mortality but also a lower risk of relapse and death from sarcoma, thus improving long-term OS [6,7]. 

Specialized centers are more likely to follow clinical practice guidelines and facilitate multidisciplinary discussions, an approach that is also associated with improved survival [8]. A MDT should include at least one surgeon, a radiologist, a pathologist, a medical oncologist, and a radiation oncologist with extensive experience in the treatment of RPS [9]. Furthermore, specialized centers have appropriate tools and facilities, such as access to pathology, molecular diagnostic tools, and a patient registry that allows quality of care assessment. Given the low prevalence of RPS and the limited benefit of therapies beyond surgery, high volume/referral centers are in an optimal position to contribute to RPS research by participating in clinical trials and national/international collaboration. 

Recommendation

In the case of a suspected RPS, patients should be referred promptly to a sarcoma expert center or reference network (IV, A).

## 4. Diagnostic Approach to RPS: Imaging and Pathology Diagnosis

### 4.1. Imaging Diagnosis

The imaging technique of choice for the diagnosis of RPS is a computed tomography (CT) scan with intravenous iodinated contrast. The administration of intravenous iodinated contrast is highly recommended because it allows the identification of areas of enhancement, not necrotic, and to obtain a biopsy with greater diagnostic accuracy [10,11,12,13]. It is important to determine whether the tumor arises from retroperitoneal soft tissues or from a retroperitoneal organ as the former are less frequent but typically malignant entities (e.g., a heterogeneous fatty mass arising from retroperitoneal tissue may be an LPS, whereas a renal fatty lesion suggests an angiomyolipoma) (Figure 1) [11,13]. Certain morphological signs allow the radiologist to deduce the origin of a retroperitoneal mass, such as the phantom organ sign, the embedded organ sign, the beak sign, and the prominent feeding artery sign [11].

A CT scan can reveal possible anatomical variants or incidental findings in the abdominal viscera. This is especially important regarding the contralateral kidney when future tumor resection implicates nephrectomy [12].

CT is also the standard method to assess tumor extension and to consider surgical planning and neoadjuvant treatment [10,13]. It is mandatory to include a chest CT to evaluate potential thoracic dissemination, particularly in cases of LMS (where up to 50% of patients may present pulmonary metastases at the moment of diagnosis) [10,12].

Magnetic resonance imaging (MRI) is a useful tool to assess the tissue composition of the tumor and to orientate its histological subtype [11,13], but it is reserved for patients with iodine allergies [12]. MRI is also strongly recommended in cases of pelvic-originated neoplasms to determine the anatomical relationships between the tumor, local viscera, and pelvic structures.

Although 18F-FDG PET-CT (positron emission tomography with 2-deoxy-2-18F-fluoro-D-glucose integrated with CT) can be used to evaluate intermediate/high-grade soft-tissue neoplasms [10], it cannot always differentiate low-grade tumors from benign lesions [12,14] and therefore has no routine role in the diagnosis or assessment of tumoral extension of RP [12] except in selected doubtful cases.

Recommendations

A CT scan is the imaging technique of choice for the diagnosis and evaluation of resection of retroperitoneal sarcomas (IV, A).MRI is also recommended to evaluate pelvic tumors (IV, A).

### 4.2. Biopsy

In patients with a suspected RPS, histological sampling should be conducted before any treatment is undertaken, except in exceptional cases where biopsy is high risk and a clear radiological diagnosis is available, e.g., well-differentiated liposarcoma (WDLPS) [15,16]. 

A percutaneous image-guided core needle biopsy is usually preferred to a surgical biopsy. The biopsy target and the needle trajectory should be determined following a thorough review of all the patient’s radiological studies and ideally after discussion by an MDT. Most retroperitoneal tumors can be percutaneously biopsied without entering the peritoneal space.

When a retroperitoneal mass is heterogeneous in the radiological examinations, the biopsy should be directed to the most “dedifferentiated” solid areas [17].

Multiple (3–4) biopsies using a 14–16 G trucut needle are recommended. A coaxial biopsy system may be used to obtain multiple tumor samples with a single percutaneous access. Percutaneous biopsy of deep retroperitoneal tumors is usually performed under CT image guidance, but large or superficial tumors can be safely biopsied using ultrasound guidance. 

If the standard safety requirements for radiological interventional procedures are fulfilled, the rate of early complications of percutaneous needle core biopsy in RPS is low, and the risk of tumor seeding also seems to be low (0.5–2%) [18,19].

In metastatic retroperitoneal sarcomas, a biopsy of the metastatic sites could be considered, as it would allow us to confirm the histological diagnosis and the advanced stage in a single procedure. 

Recommendations

Before any treatment for a suspected RPS, a core needle biopsy should be performed (IV, A).It should be directed to the most “dedifferentiated” solid areas (IV, A).In RPS with metastases, a biopsy of the metastatic sites could be considered to reach a histological diagnosis if they are more easily accessible (V, B).

### 4.3. Pathological Diagnosis of Soft Tissue Sarcomas—Indication of Molecular Studies

The pathology report of a trucut biopsy should include the histological type, or if this is not possible, at least establish the morphologic category (spindle cell, myxoid, pleomorphic, round cell, etc.), the histologic grade and the results of the complementary studies performed (immunohistochemistry (IHC) and/or molecular biology).

From the resection specimen, the pathologist must provide the following information:A macroscopic description: measurements of the surgical specimen, type of surgical specimen, and identification of the tissues and organs included.Description of the tumor: size, appearance, location, presence of necrosis, and invasion of neighboring structures.Resection margins: the distance of the tumor to the margins should be measured and those that are less than 2 cm should be specified. It should be indicated whether the margin is formed by fascia, visceral, adventitial, or periosteal tissue.Presence and description of satellite nodules.Lymph nodes: although lymph node involvement is rare in STS (except for rhabdomyosarcoma (RMS), angiosarcoma, or epithelioid sarcoma), the status of any lymph nodes present should be included.Any additional techniques performed should be reported: IHC, reverse transcriptase-polymerase chain reaction (RT-PCR), next generation sequencing (NGS), multiplex ligation-dependent probe amplification (MLPA), fluorescence in situ hybridization (FISH), and their results.

The most common subtypes in this location are LPS and LMS [20]. Other less common subtypes occurring in the retroperitoneum are SFT, MPNST, undifferentiated pleomorphic sarcoma (UPS), intimal sarcoma (IS), synovial sarcoma (SS), perivascular epithelioid cell tumor (PEComa), and undifferentiated small round cell sarcoma. 

Retroperitoneal LPS is subclassified into four histologic types: WDLPS, DDLPS, pleomorphic (PL), and myxoid LPS (MXLPS). The latter two entities are extremely rare, and it is not unusual that even large series have no cases. 

Because DD-LPS can present a varied histomorphology (spindle cell, pleomorphic, myxoid, small round cell, or epithelioid), the use of IHC techniques (antibodies against MDM2, CDK4, and p16), at least in doubtful cases, is highly recommended and helpful in the RPS diagnosis. We should consider that RPS other than DD-LPS may be positive for MDM2/CDK4 (IS and MPNST). Appropriate IHC techniques, including neural and myoid markers, should be performed to rule out these possibilities (Table 2).

Detection of MDM2 amplification by FISH is currently the gold standard for the diagnosis of WD/DDLPS. It is particularly useful in the following situations: (1) to confirm the diagnosis of WDLPS in an adipocytic tumor with minimal cytologic atypia; (2) to establish the diagnosis of DDLPS in a relatively non-descript spindle cell or pleomorphic retroperitoneal sarcoma, and (3) to classify a pleomorphic or myxoid adipocytic sarcoma such as DDLPS with homologous lipoblastic differentiation that could be mistaken for pleomorphic or myxoid liposarcoma (Figure 2).

The diagnosis of primary retroperitoneal MXLPS should be made with caution because such cases represent either metastatic disease or stromal changes in WDLPS/DDLPS. In absence of MDM2 amplification, the demonstration of FUS translocation is useful for the diagnosis.

Smooth muscle tumors usually have a spindle-shaped morphology. They are positive for myoid differentiation and need to be positive for at least two markers, including smooth muscle actin (SMA), desmin, H-Caldesmon, calponin, and smooth muscle myosin heavy chain. This histomorphology of LMS bears a certain semblance to other entities such as myoid differentiation in DD-LPS, gastrointestinal stromal tumor (GIST), PEComa, and IS. Consequently, a judicious panel of immunohistochemical markers is necessary to ensure correct classification, including SMA, desmin, H-Caldesmon, MDM2, CDK4, HMB45, CD117, and DOG-1. 

IHC of MPNSTs may be diagnostically useful in some cases. Half of the cases retain S100 or SOX10 expression. However, this may be only patchy or focal in spindle cell morphology. Loss of H3K27me3 expression occurs in approximately 50% of cases. Nevertheless, this usually occurs in high-grade neoplasms. 

RMS is even less frequent in the retroperitoneum. Therefore, if a primary RMS is diagnosed in this location the possibility of a MPNST (Triton Tumor) or a DD-LPS with heterologous elements should be considered.

SS may be monophasic, biphasic, or poorly differentiated. In the immunohistochemical study, EMA expression is more frequent than cytokeratin expression, and focal expression of S100 may be detected (40% of SSs). The vast majority of SS are positive for CD99 and for the transcriptional corepressor TLE-1 (transducin-like enhancer of Split 1) with strong nuclear expression. Classically, the diagnosis of synovial sarcoma is confirmed by demonstrating the SS18 gene translocation by FISH. Antibodies specific to the SS18-SSX fusion (E9X9V, designed for the breakpoint) and SSX (E5A2C, designed for the C-terminus of SSX), which typically exhibit nuclear staining, are good surrogate markers of the characteristic SS translocation (sensitivity (95%) and specificity (100%)).

In SFT, STAT-6 detection by IHC is the most useful diagnostic marker, with nuclear expression identified in more than 95% of cases. 

It is important to remember that benign soft tissue tumors with a retroperitoneal location should also be considered, such as angiomyolipoma, lipoma, hibernoma, leiomyoma, schwannomas, and neurofibromas.

For the evaluation of the pathologic response in those RPS that have received neoadjuvant treatment, it is advisable to follow the recommendations of the European Organization for Research and Treatment of Cancer-Soft Tissue and Bone Sarcoma Group (EORTC-STBSG). By establishing uniform criteria and methodologies, these guidelines provide a framework for pathologists and clinicians to consistently assess treatment response in soft tissue sarcomas, facilitating effective communication and comparisons of outcomes between different treatment centers and research studies [22].

Recommendations

Detection of MDM2 amplification by FISH is currently the gold standard for the diagnosis of WD/DDLPS (I, A).Molecular testing has no diagnostic role in leiomyosarcoma, SFT, or MPNST (I, A).

## 5. Preoperative Functional Assessment

To achieve better postoperative outcomes and avoid the development of potential complications the patient’s preoperative evaluation should focus on assessing comorbidities, performance status (PS), nutrition, kidney function, and the American Society of Anesthesiologists (ASA) Physical Status Classification System. In a preoperative clinical visit, the patient’s clinical symptoms (neurological and vascular-related symptoms) and previous comorbidities should be evaluated. When multivisceral resections are needed, the kidney is one of the most frequently resected organs. Therefore, renal function should be evaluated using creatinine and the glomerular filtration rate (GFR) in blood tests, as well as creatinine clearance in urine [23].

Protein energetic malnutrition (PEM) was documented for the first time in RPS patients who underwent surgery. As described in the Italian Society for Enteral and Parenteral Nutrition (SINPE) guidelines, PEM includes upper arm circumference (cm), weight loss >5% of body weight, lymphocytic count (number/mm^3^), prealbumin (mg/dL) and transferrin (mg/dL) levels [24]. As the potentially significant tumor mass could alter body weight, the body mass index (BMI) was ignored. Nutritional support should be included in enhanced recovery programs after surgery programs [25,26]. 

Recommendations

Perioperative assessment and support are recommended when surgical resection is planned (IV, A).Renal function and nutritional status should be evaluated during surgical planning (IV, A).

## 6. Staging and Risk Assessment

Nomograms are statistical tools designed to predict an individual patient’s oncologic outcome and are based on the simultaneous effect of several prognostic factors [27]. Correct prediction of relapse risk and oncologic outcomes is essential in clinical decision making, and can help select patients for clinical trials [25].

In the setting of RPS, the TNM staging system and non-specific nomograms have a limited ability to predict prognosis [25]. Several RPS-specific nomograms have been developed. However, they are all based on postoperative variables and cannot, therefore, be used in the preoperative setting or for patients with metastatic or unresectable RPS [25]. 

In patients with primary RPS, several nomograms, such as the multi-institutional Gronchi [28] and Callegaro’s dynamic nomogram [29], allow the calculation of OS and disease-free survival (DFS), while others calculate local and distant recurrence [30,31,32]. For patients with recurrent RPS, we can use Raut’s multicenter (TARPSWG) nomogram to calculate OS and DFS [33] (Table 3).

Recommendation

The use of nomograms as predictive tools for survival and risk of relapse may be useful in adjuvant treatment decision-making and patient selection for clinical trials (III, C).

## 7. Treatment of Resectable Localized Disease 

### 7.1. The Primary Surgical Approach in Localized Disease—Adapting the Surgical Approach to the Histological Subtype

RPS surgery should be performed in referral centers by surgeons experienced in abdominal, retroperitoneal, and pelvic surgery and management of vascular and genitourinary techniques [34,35,36,37]. 

The various histologic types of RPS have a specific risk of locoregional and distant recurrence, providing an additional basis for histology-guided surgery [31,32,33,34].

In LPS, local recurrence (LR) is the main cause of relapse and disease-related mortality, so it is especially important to obtain free surgical margins in the initial resection of the primary tumor. Intraoperative assessment of margins is difficult, especially in well-differentiated tumors, and preservation of potentially infiltrating organs increases the risk of LR. To avoid LR, it is recommended to remove all the fatty tissue in the affected retroperitoneal space “en bloc”, adopting a policy of “liberal organ resection” [31,32,33,34]. Left tumors should be resected en bloc with the left colon and the left kidney. The tail of the pancreas and the spleen should be included in the resection, if they are englobed or in intimate contact with the tumor, which would prevent their resection with an R0 margin, assuming an increase in morbidity. For right-sided tumors, the right colon and kidney should be included in the resection. 

LMS usually originates from great blood vessels such as the inferior vena cava, renal, gonadal, or iliac veins. These tumors have a high incidence of distant metastasis and a low incidence of LR. In such cases, adjacent organs should be preserved if they are not directly adherent to, or invaded by the tumor. Major vascular resections with or without reconstruction may be necessary [4,38].

SFT generally presents a low risk of LR, so the goal should be complete resection with negative margins [4,39].

Sarcomas originating in the psoas are usually undifferentiated/unclassified sarcomas and may extend below the inguinal ligament into the thigh, although they are usually separated from the retroperitoneum by the psoas fascia. The goal is to remove the tumor and muscle “en bloc” with the surrounding fascia, sparing adjacent nerves, vessels, and viscera if unaffected [4,34].

For MPNST arising from retroperitoneal nerves, resection should be complete, with negative microscopic margins. Locally advanced MPNST of the retroperitoneum has a poor prognosis and complete resection can be difficult. Surgical judgment should be used to determine the resection of adjacent major neurovascular structures [31,34].

Technical criteria for unresectability are involvement of the superior mesenteric artery, aorta, celiac trunk and/or portal vein, bone involvement, growth into the medullary canal, an invasive extension of the retrohepatic inferior vena cava, leiomyosarcoma in the right atrium, and infiltration of multiple major organs such as the liver, pancreas and/or major vessels [31,34].

In the surgical treatment of primary RPS, preservation of specific organs (e.g., kidney, pancreas, spleen, and colon) should be individualized, taking into account the biology of the tumor, its extent, and the patient’s characteristics (V, A) [31,32,33,34].

Recommendations

RPS surgery should be performed in referral centers by experienced surgeons (III, A).In LPS, it is recommended to remove all adipose tissue in the affected retroperitoneal space “en bloc” following a policy of “liberal organ resection” (III, A).In “non-liposarcoma” LPS, surgery should attempt to achieve a macroscopically complete resection in a single piece encompassing the tumor and contiguous affected organs (III, A).The decision to preserve specific organs should be individualized, taking into account the biology of the tumor, its extent, and the patient’s characteristics (V, A).

### 7.2. Management after Simple Excision (with Residual Macroscopic Disease) 

Incomplete resections in RPS are associated with a significantly higher risk of local and distant recurrence and poorer survival [40,41,42,43]. In addition, such surgery may cause unnecessary morbidity and mortality as it has no demonstrable beneficial effect on survival compared to that in patients with unresectable disease [1,2,34,37,38,39,40].

Low-grade LPS is the only histologic subtype in which cytoreductive procedures can improve survival and help improve symptoms [2,40].

The optimal treatment strategy for patients with residual disease after inadequate primary RPS surgery is unknown [44,45].

If curative resection is attempted, the intention should be to reproduce what would ideally have been done for the primary RPS in its original state, since the possibility of disease control may be thereby increased, despite previous operative interference [34,38].

If the first surgical intervention consisted of a simple excision that left macroscopic residual disease (identified on cross-sectional imaging shortly thereafter), careful consideration should be given to the timing of any subsequent surgical intervention to attempt a curative resection. An observation period is appropriate to rule out the multifocal spread of high-grade disease at the time of the previous intervention [34].

In WDLPS, initial surveillance may be considered an option, reserving resection for significant growth or the appearance of a DD component [34].

Recommendations

Unplanned, grossly incomplete resection should be avoided (III, A).Patients with RPS undergoing inadequate primary surgery should be referred to specialized sarcoma centers, and complete surgery may be considered (IV, A).The extent of resection should be as required to achieve complete gross resection (IV, B).

### 7.3. Preoperative or Postoperative Radiotherapy

Radiotherapy (RT) treatment recommendations for RPS have traditionally been based on those established for extremity sarcoma. Perioperative RT in RPS has been associated with improved OS compared to surgery alone in retrospective studies [46].

The only published prospective randomized phase III study (EORTC-62092: STRASS), comparing surgery alone versus preoperative RT followed by surgery in patients with newly diagnosed resectable RPS failed to show abdominal recurrence-free survival (ARFS) (primary endpoint) or OS benefit [47]. In a posthoc analysis, the subgroups of LPS and low-grade RPS showed a trend toward increased ARFS at 3 years among those treated with RT, 65.2% (95% CI 54.5–74.0) in the surgery group vs. 75.7% (65.6–83.2) in the preoperative RT and surgery group, (HR 0.62, 95% CI 0.38–1.02). However, for the subgroup of LMS and high-grade DDLPS preoperative RT did not show this benefit. These results should be taken with caution, nevertheless, as most patients did not receive the standard dose of RT. Subsequently, data from patients with resectable primary RPS who received preoperative RT within the STRASS trial and those who received the same treatment outside the trial (STREXIT) among ten STRASS recruiting centers have been compared. This study confirms previous results and suggests a possible survival benefit for patients who received preoperative RT, but as this is a retrospective study, caution should be exercised in interpreting the results due to inherent biases [48].

The standard dose and fractionation for preoperative RT Is 45–50 Gy in daily fractions of 1.8 Gy or 2 Gy [49]. Whenever possible, intensity-modulated RT (IMRT) should be employed to optimize the therapeutic index by further reducing the dose to normal tissues. As intensity-modulated protons (IMPT) in the treatment of RPS enable a dose reduction in critical organs such as the kidney (V15 16.4%) and the small bowel (V45 6.3%) with dose conformities comparable to IMRT, phase I studies have suggested the possibility of dose escalation in areas where an adequate surgical margin is difficult to achieve [50]. The time from RT to surgery should be 4 to 8 weeks.

The use of postoperative RT in RPS is limited by its high toxicity due to the need to irradiate large volumes after resection, in addition to inter-fractional and intrafractional movements, and low intestinal tolerance. Although adjuvant RT is not generally recommended in RPS because of its high late toxicity (estimated to be between 5% and 40% for doses between 50 and 60 Gy), most authors agree that it may help improve local control [51]. 

Institutions that have used intraoperative radiotherapy (IORT) after surgery have reported a decrease in the risk of areas of residual microscopic disease and improved local disease control, but no clear benefits in OS [52]. The combination of preoperative RT, surgery, and IORT (10–20 Gy) has achieved higher local control rates (60–83%) with acceptable toxicities [53]. Major nerves (risk of neuropathy if dose > 12.5 Gy), gastrointestinal structures, and ureters should be removed from the IORT site whenever possible to avoid dose-limiting toxicities.

Recommendations

Preoperative RT should not be routinely performed in resectable RPS (I, C).Neoadjuvant RT may be considered in primary low/intermediate grade retroperitoneal LPS (II, B).Postoperative RT is generally discouraged due to the high risk of toxicity (IV, D).As a dose-escalation technique, IORT after maximal stress surgery may reduce the risk of microscopic residual disease areas, improving local disease control (II, B).

### 7.4. Adjuvant/Neoadjuvant Chemotherapy 

To date, no randomized trials in RPS have compared neoadjuvant or adjuvant chemotherapy (CT) with surgery alone. Data from adjuvant/neoadjuvant studies in high-risk extremity STS cannot be extrapolated to RPS [34]. Although the results of the study by Gronchi et al. support the use of neoadjuvant chemotherapy in high-risk soft tissue sarcomas, it should be noted that the retroperitoneum often presents tumors that are potentially less sensitive to such treatment [54].

In a cohort study with 169 patients neoadjuvant CT was associated with lower overall survival [55]. Adjuvant CT has also been associated with worse survival in a meta-analysis of 15 clinical trials [56]. However, these studies included a non-selected RPS population, whereas CT could have a role in specific histologies, such as grade 3 DDLPS and LMS, that are more chemosensitive and associated with a higher incidence of distant metastasis [57]. 

A recent publication of the TARPSWG analyzed the efficacy of neoadjuvant CT in 158 patients with retroperitoneal STS. Although there was significant heterogeneity, most regimens were anthracycline-based. Globally, 23% of patients achieved a partial response (PR), 56% had stable disease (SD), and 21% showed tumor progression (PD). Patients with grade 3 DDLPS had a similar overall response rate (ORR) with an anthracycline + ifosfamide regimen (23%), and with another regimen (25%). However, patients with LMS had a higher ORR with an anthracycline + dacarbazine regimen (37%), than with another regimen (16%) (*p* = 0.17). Globally, those who experienced PD before surgery had worse OS [58].

Although rare in this location, other histotypes, such as MPNST or SS, could also benefit from neoadjuvant CT. Moreover, preoperative administration of certain targeted therapies could be considered in other infrequent subtypes in which surgery with adequate margins cannot be performed initially (e.g., crizotinib in ALK+ inflammatory myofibroblastic tumor) given the high ORR achieved in advanced disease [59].

Currently, the phase III STRASS-2 trial is recruiting patients to assess the role of neoadjuvant CT versus resection alone in high-risk RPS in two cohorts: (1) grade 3 DDLPS, with the doxorubicin-ifosfamide regimen in the CT arm, and (2) grade 2–3 LMS, compared to doxorubicin-dacarbazine in the CT arm. In contrast, cohort C of the phase II TRASTS trial is enrolling patients with retroperitoneal high-grade LMS, DD-LPS, or pleomorphic LPS to explore the efficacy of concomitant trabectedin plus low-dose RT. This combination has shown synergistic activity in a metastatic STS cohort, with a centrally assessed ORR of 60% [60]. Given the scant evidence of the role of neoadjuvant CT, it is highly recommended to include patients with high-risk RPS in clinical trials is highly recommended.

Recommendations

Neoadjuvant or adjuvant CT is not recommended in a non-selected RPS population (II, C).However, preoperative CT could be considered in selected patients, mainly in unresectable or borderline resectable cases with grade 3 DDLPS (with anthracycline-ifosfamide) and LMS (with anthracycline-dacarbazine) (IV, C).Preoperative targeted therapy could be considered for some specific, rare histotypes (III, C).

## 8. Treatment of Local Recurrence

Although primary extended surgery with complete resection is the best treatment to avoid LR, it will not prevent relapse in a high percentage of patients treated for RPS. Among the various histological subtypes, LPS has the highest rates of LR (30–50%), constituting the main cause of disease-related mortality. LR is a challenging scenario that should always be evaluated in an MDT [41].

Surgery should be the first option in the presence of a resectable LR as it offers the possibility of complete surgical remission and, in selected cases, a possible cure [38,61]. However, successful complete resection is a difficult goal given the high risk of progressive recurrence and associated surgical morbidity. Specific prognostic nomograms for recurrent RPS can help in the therapeutic decision [33].

There are several possible scenarios in the event of a local recurrent RPS [41,42,46]:-If the primary surgery was a marginal surgery in a non-reference center without an expert MDT, extended en bloc resection should be considered in cases with isolated recurrences (especially in LPS and above all WDLPS) and tumor growth rates < 1 cm/month.-If the primary surgery was extended and complete, surgery should be offered if macroscopic resection is possible, with favorable histology and a previous disease-free interval >1 year.-In the particular case of WDLPS, it is advisable to monitor the initial evolution of the recurrence and avoid very early intervention [62].

Recurrent RPS (isolated or multiple lesions) characterized by previous tumor rupture, multifocality, high histological grade, or short disease-free interval are indicative of poor prognosis. In such cases, adequate patient selection by the MDT is required. Surgery should be limited to resection of all tumor lesions and as conservative as possible [41,63].

Recommendations

Surgical treatment decided by an experienced MDT is the treatment of choice for the first resectable LR (IV, A).The resection of successive recurrences, after patient selection (based on histology, morbidity, and free interval) should be complete but not extended to non-infiltrated contiguous organs (IV, B) [63,64].Patients with isolated LR that have not been irradiated previously and patients with well or moderately differentiated liposarcomas could be considered for preoperative RT (IV, B) [41,65].Neoadjuvant therapy should be considered, especially in recurrent RPS with high-grade, borderline complete resection, a short disease-free interval, or high surgical morbidity (V, B) [41].

## 9. Treatment of Localized Unresectable Disease 

Approximately 10–25% of non-metastatic RPS are considered inoperable in referral sarcoma centers [34]. Although there are no well-established criteria, there is considerable unanimity to consider the involvement of the celiac-mesenteric vessels and other critical vascular structures as the main technical criteria for unresectability [66]. The other main reason for ruling out surgery in these patients is poor PS and comorbidity [67].

Patients with technically unresectable RPS but an acceptable PS and patients with metastasis should be treated with CT. CT could be guided by tumor histology, especially in patients with borderline resectable disease in whom it is important to obtain a response. In LPS, the WD component is usually resistant to CT, and response is only seen in the DD component [68].

When an objective response is achieved in a patient with RPS, the possibility of surgery should be reassessed. If surgery is definitively ruled out, consolidation with RT could be an option if not administered earlier. 

There is no consensus on the optimal RT for patients with unresectable disease, either at diagnosis or recurrence. Due to the presence of healthy tissues, it is necessary to use advanced RT techniques such as intensity-modulated image-guided radiotherapy (IG-IMRT), IORT, brachytherapy, and even protons and carbon ions. Evidence in this group is provided by a systematic review of 11 retrospective studies and case series where local control rates with exclusive RT for up to 24 months using high doses (median 63 Gy) have been reported [69]. A study of 14 patients showed PR in 6 patients, SD in 7 patients, and progression despite RT in one patient. The median local control time and OS were 27.7 and 41.5 months, respectively.

Three of the series included in the review used brachytherapy with long-lasting responses but observed toxicity, mainly involving neuropathy, hydronephrosis, fistulas, and abscesses. The use of expanders is thus recommended to avoid complications [70]. The use of protons and carbon ions has also provided satisfactory results, especially in locations with difficult surgical access, at a dose of 63 Gy [71,72].

Other approaches to this disease could be the use of concomitant CT and RT, although this association has only been described in the context of STS of the extremities [73].

Finally, patients with poor PS or serious comorbidity are candidates for less aggressive palliative CT, palliative RT, or best supportive care (BSC) [74].

Recommendations

Patients with technically unresectable RPS with an acceptable PS, and especially high-grade, and sensitive histologies should be treated with CT (IV, B). In the case of an objective response, surgical possibilities should be reconsidered (IV, B).In unresectable RPS with chemoresistant histologies (e.g., WDLPS) and in patients who are not candidates for CT (PS or comorbidity), RT may be an effective option (i.e., SFT) (IV, B).Palliative RT, less aggressive CT, and BSC are options that should be considered to alleviate symptoms in patients with poor PS or severe comorbidities (IV, B).

## 10. Treatment of Advanced Disease

### 10.1. Surgical Treatment of Advanced Disease

Advanced disease includes distant involvement, with lung and liver being the most frequent sites, and multifocal intra-abdominal involvement or sarcomatosis. Despite multidisciplinary management, the prognosis of metastatic RPS is poor, with an OS of 16 months, and only 5% of patients are alive at 5 years. In most cases, CT is the first treatment option, but in selected cases, complete surgical eradication of metastatic RPS disease may reduce symptoms, improve survival and increase the likelihood of re-resection in the face of further recurrence [4,75]. It is therefore essential to properly establish surgical indications that will improve patient prognosis [1,2,76]. Such indications are based on the controllable primary tumor, the histologic subtype, its chemosensitivity, the number of metastatic lesions (oligometastatic disease), the patient’s functional status, and comorbidity [2]. 

When sarcoma pulmonary metastases are detected, their resection, through the thoracotomy approach, is the best therapeutic option (40–50% OS at 3 years vs. 10% in non-resected patients). In metachronous lung metastases, the Bethesda criteria are useful to evaluate candidates for surgical resection. They take the following factors into consideration: double tumor growth in >20 days, number of metastases ≤ 4, a free-interval of disease > 12 months, and complete tumor resection [77]. If pulmonary metastases are synchronous with other locations, surgery is considered if stabilization is achieved with previous CT [78]. In selective patients, minimally invasive techniques (thoracoscopic resection, RF/microwave ablation) should be considered as an alternative [79]. 

In well-selected patients, exercision of metachronous hepatic metastases from an RPS can improve OS [80]. Synchronous liver metastases should be treated with chemotherapy first, and surgery should be offered for responding metastases in selected patients. In these cases, surgical removal can be synchronous or deferred, following a sequential order in the removal of all existing lesions in liver metastatic disease [76].

Intraperitoneal dissemination is usually multifocal, of variable size, and often caused by a tumor rupture in a previous surgery. In these cases, surgery is generally incomplete and should be considered inadequate. Surgical resection should be considered in patients with favorable biology (low grade tumor, low volume in number and size, and long disease-free interval) and when complete resection is expected. Cytoreductive surgery together with intraperitoneal hyperthermic CT (HIPEC) in peritoneal sarcomatosis is associated with high toxicity and no clear benefit, so it is only acceptable within clinical trials [36].

Recommendations

In metastatic PRS disease, prior evaluation by a MDT is essential to properly establish the surgical indications (IV, B).Surgery in oligometastatic disease may be considered in selected patients with good PS and favorable tumor biology (DFS greater than 12 months) or prolonged control (ORR or SD) of disease with systemic CT therapy (IV, B).

### 10.2. Radiotherapy Treatment in Advanced Disease

In metastatic disease, the approach should be personalized and agreed upon by a tumor board. Extracranial stereotactic radiotherapy (SBRT) is a non-surgical alternative in oligometastatic patients, with a long interval between the end of the first treatment and the appearance of metastases [81]. Palliative RT is a useful option for managing disease-related symptoms such as pain, bleeding, or spinal cord compression [69].

Recommendations

SBRT should be considered an option in the management of oligometastatic disease in patients who are not candidates for surgical management (IV, B).RT may be used for the purpose of relieving RPS-related symptoms. (IV, A).

### 10.3. Systemic Treatment of Advanced Disease

In the case of advanced (irresectable or metastatic) RPS, systemic therapy is the cornerstone of treatment, aiming to control the disease and symptoms. The same principles of systemic therapy in advanced disease for STS in other locations apply to RPS. Inclusion in clinical trials should always be considered in this population. Anthracycline-based CT is the standard first-line treatment, although sensitivity is highly variable across the various histologic subtypes [82,83]. Although the use of anthracycline-based combinations is not associated with increased survival, it is associated with higher objective response rates. Therefore, they may be considered in selected patients in whom a dimensional tumor response could aid symptom control or facilitate surgical salvage [84]. 

In advanced LMS, combinations of anthracyclines with trabectedin [85] or dacarbazine (DTIC) [86] have shown activity and are alternatives to combinations with ifosfamide, a drug that has not shown relevant activity in this histologic subtype [87]. Beyond anthracyclines, there are several active drugs that could be administered in advanced disease. Eribulin is approved for the therapy of advanced LPS and has been shown advantages in terms of OS when compared to DTIC [88]. Trabectedin is approved in STS pretreated with anthracyclines and ifosfamide, and has shown to be superior in terms of progression-free survival (PFS) compared to dacarbazine [89]. It is especially active in L- sarcomas and translocation-related sarcomas, although there is evidence of activity in other histologic subtypes. The combination of trabectedin and palliative RT is feasible and has shown activity, and can be an option in patients with symptomatic lesions needing dimensional shrinkage (III, B) [55]. Pazopanib is approved in pretreated advanced STS (except for LPS), based on its superiority over BSC [90]. High-dose ifosfamide in a continuous infusion (14 g/m^2^ in 14 days) has shown activity in LPS and can be a second-line option, even in patients progressing to standard doses of ifosfamide [91]. Gemcitabine (GZT) combinations are second-line options, especially active in LMS. The combination of GZT and dacarbazine (DTIC) was superior to DTIC [92], whereas there are contradictory data on the superiority of GZT plus docetaxel when compared to GZT alone, and its toxicity profile is worse than GZT and DTIC [93,94]. It is relevant to assess the radiological response in RPS. Responses to systemic therapy can be non-dimensional, especially with drugs such as trabectedin and pazopanib. In addition, the response can be heterogeneous among the WD and DD components of a retroperitoneal LPS, and there is no clear consensus regarding the extension of the disease to be considered for response evaluation in this scenario (all the disease burden versus the DD component). 

Recommendations

In low-grade RPS, especially in asymptomatic patients, active surveillance may be a good option (IV, C).Anthracycline-based chemotherapy is the standard first-line treatment of advanced disease (II, A). Anthracycline-based combinations (II, A) can be evaluated in fit patients, when surgical salvage is the goal, and in patients in whom a dimensional response might improve symptoms.Several second-line and subsequent treatments are available for the treatment of patients after progression or in those who are ineligible for first-line, and the decision is based on histology, toxicity profile, and patient preference (IV, C):Although there is additional evidence of trabectedin activity in L-sarcomas (I, A) it can be considered in the treatment of all sarcoma subtypes (III, B).Pazopanib is indicated in the treatment of non-LPS (II, A).Eribulin is an alternative in the treatment of LPS (I, A).GZT combinations, preferably with DTIC, due to a better tolerability profile, are an especially useful alternative in LMS (II, B).High-dose ifosfamide is an option, particularly in synovial sarcoma (III, B).Inclusion in clinical trials should always be considered in this situation (IV, B).

## 11. New and Upcoming Therapies

As RPS encompasses a heterogeneous group of diseases with varying clinical behavior and prognosis, the development of new strategies of treatment is often challenging. Several drugs are currently under development for advanced and recurrent diseases, but most are still in early phase trials. Among these drugs, MDM2 inhibitors seem to be a promising approach for LPS. They show an ORR between 5–10% and high rates of stabilizations with a favorable toxicity profile [95]. Other drugs tested in LPS are CDK4/6 inhibitors and selinexor (XPO1 inhibitor), both with modest results as monotherapy treatments [96,97]. A phase 2 trial with cabazitaxel in DD-LPS has shown interesting results with a disease control rate of 68% and median PFS of 21.6 months [98]. For other tumor subtypes such as LMS, the addition of PARP inhibitors to CT is being explored [99].

The association of CT plus regional hyperthermia has also been tested in an EORTC randomized clinical trial in 329 patients with localized high-risk STS. The addition of regional hyperthermia to CT was associated with increased OS and local control [100].

There are no randomized clinical trials studying the role of hyperthermic intraperitoneal chemotherapy (HIPEC) in peritoneal sarcomatosis after cytoreductive surgery (CRS). A systematic review and meta-analysis describe that this approach may improve outcomes in selected patients, particularly those with a low tumor burden and disease amenable to complete cytoreduction [101]. Although more research is needed to confirm these findings, conducting a HIPEC clinical trial would require the unification of histological criteria, radical resection criteria, and a multi-institutional approach to achieve an adequate and homogeneous group of patients.

Immunotherapy alone or in combination has shown to be effective in some subtypes of RPS, such as DD-LPS and UPS. New combinations with novel agents such as trabectedin and cabozantinib, and biomarkers of response are being explored [102]. 

A better understanding of the molecular characteristics and the tumor microenvironment will help us develop new targeted therapies with better outcomes. The incorporation of tumor sequencing and other -omics techniques into clinical practice will be crucial to this end. It will also allow us to identify biomarker agnostics for targeted therapy, such as NTRK fusions or MET amplification [103]. 

Recommendations

Inclusion in clinical trials for advanced disease patients is highly recommended (V, A).NGS and other -omics are needed to increase knowledge, to select patients for clinical trials, and identify potential driven treatments (V, A).

## 12. Follow-Up

Despite curative resection, locoregional recurrence is common in up to 30% of patients and accounts for 75% of RPS-related deaths [104,105]. Approximately 9% of LR and 6% of distant recurrences occur after 5 years after surgery [52,106]. Some patients should undergo long-term follow-up [1] (Table 4).

Recommendations

Postoperative follow-up of patients at high/intermediate risk of recurrence should be performed with thoracic and abdominopelvic CT every 3–4 months for the first 2–3 years, every 6 months for the next 3 years, and once a year thereafter (V, B).Patients at low risk of recurrence can be followed with abdominopelvic CT and chest X-ray every 4–6 months for the first 3–5 years, then once a year thereafter (V, B).Long-term follow-up of patients beyond 5–10 years is recommended (IV, B).

## 13. Conclusions

PRS, a rare disease, requires specialized treatment in high-volume referral centers with MDT. Recent advances in understanding the biological diversity of PRRS have paved the way for personalized histology-based therapeutic approaches. These encompass surgical and non-surgical interventions, such as radiotherapy and chemotherapy. This comprehensive review presents the GEIS recommendations, providing clear and pragmatic directions for the diagnosis, treatment, and monitoring of RPS patients.

## Figures and Tables

**Figure 1 cancers-15-03194-f001:**
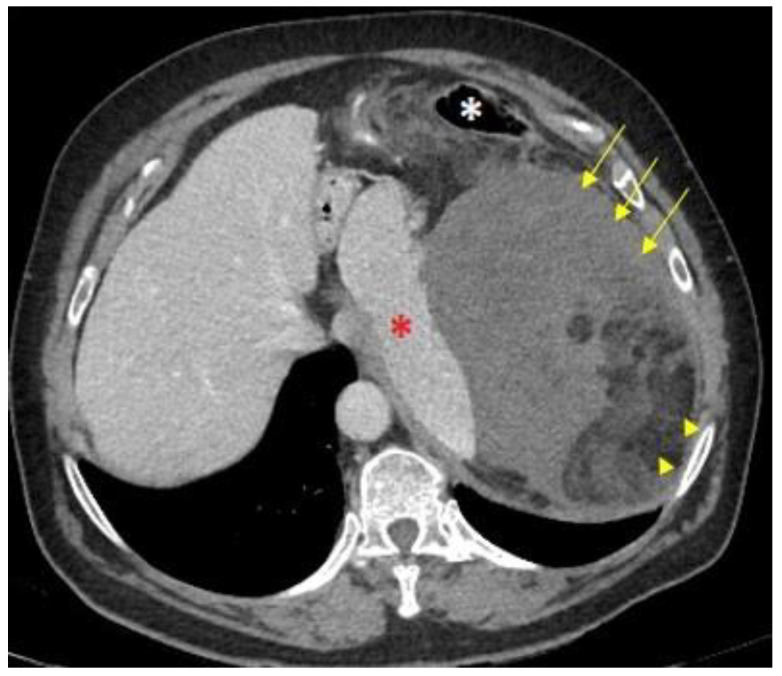
CT Morphological signs of DD-LPS: An axial computed tomography (CT) scan shows a giant mass in the left hypochondrium. The medial displacement of the spleen (red asterisk) and the anteromedial position of the descending colon (white asterisk) indicate it is a retroperitoneal-originated neoplasm. The mix of macroscopic fat (arrowheads) and soft-tissue components (arrows) suggests a dedifferentiated liposarcoma.

**Figure 2 cancers-15-03194-f002:**
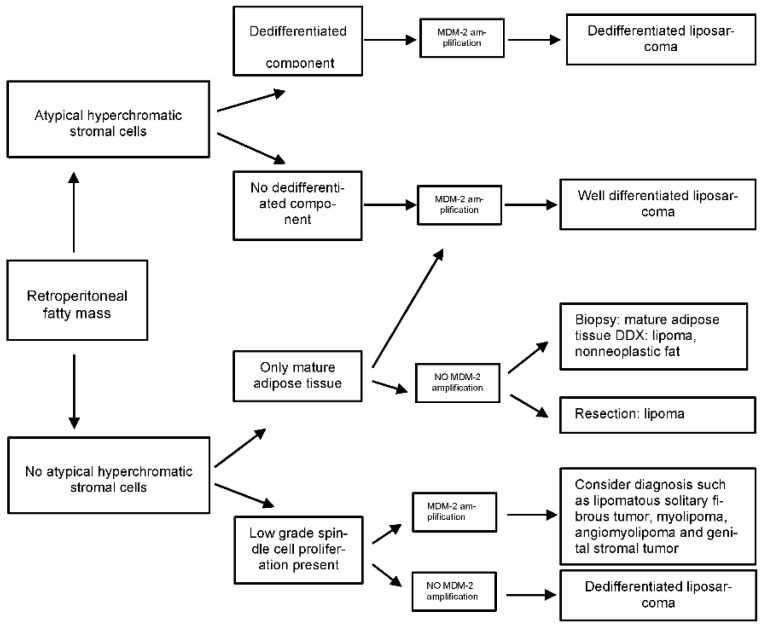
Algorithm for the evaluation of a fatty retroperitoneal mass. Algorithm for evaluation of a “fatty” retroperitoneal mass adapted from Karen J Fritchie [21].

**Table 1 cancers-15-03194-t001:** Levels of evidence (I to V) and grades of recommendation (A to C).

Levels of Evidence	
I	Evidence from at least one large, randomized, controlled trial of good methodological quality (low potential for bias), or meta-analyses of well-conducted randomized trials without heterogeneity
II	Small randomized trials or large randomized trials with a suspicion of bias (lower methodological quality), or meta-analyses of such trials or of trials with demonstrated heterogeneity
III	Prospective cohort studies
IV	Retrospective cohort studies or case-control studies
V	Studies without a control group, case reports, and experts’ opinions
Grades of recommendation	
A	Strong evidence for efficacy with a substantial clinical benefit, strongly recommended
B	Strong or moderate evidence for efficacy but with a limited clinical benefit, generally recommended
C	Insufficient evidence for efficacy or benefit does not outweigh the risk or the disadvantages (adverse events, costs…), optional

**Table 2 cancers-15-03194-t002:** Recommendation of Immunohistochemistry in RPS.

Recommendation for Immunohistochemistry in Adipocytic Tumors or Tumor with Fatty Areas in Retroperitoneum	
MDM2/CDK4	To distinguish between benign and malignant adipocytic tumors or to subclassify LPS
HMB-45/STAT-6	Angiomyolipoma or SFT
MYOGENIN	Allows recognition of rhabdomioblastic differentiation in DD-LPS
Immunohistochemistry techniques to consider in retroperitoneal fusocellular tumors	
MDM2/CDK4	LPS (DD-LPS or WD-LPS), IS, MPNST
SMA/Desmin/H-Caldesmon	LMS or IS
CD34/STAT6	SFT
S100/SOX10/H3K27me3	MPNST/neural tumor
CKIT/DOG-1	GIST
SS18-SSX/TLE-1/EMA	SS
HMB-45/MELAN-A	PEComa or metastatic melanoma
MYOGENIN	RMS or rhabdomyoblastic differentiation in other STS

**Table 3 cancers-15-03194-t003:** Nomograms for patients with RPS.

Author, Center (Year)	Selection Criteria	Timeframe	Number of Patients	Predicted Outcomes	Covariates Included	External Validation	Concordance Index	Observations
Gronchi, INT, UCLA, MDACC, (2013) [28]	Primary Localized Resected	1999–2009	523	7-year OS	Grade Size Histology Age Multifocality Extent of resection	Yes	0.74	Digital version available in the Sarculator app (www.sarculator.com)
			475	7-year DFS	Grade Size Histology Multifocality	Yes	0.71	
Tan, MSKCC (2016) [30]	Primary Localized Resected	1982–2010	632	3, 5, 10-year DSD	Histology Extent of resection Number organs resected Size Radiation associated	Yes	0.71 (0.66–0.74)	Available web-based calculator
			574	3, 5, 10-year LR rate	Histology Size Age Resection Location Vascular resection Number organs resected	No	0.71 (0.67–0.75)	
			632	3, 5, 10-year DR rate	Histology Number organs resected Size Radiation associated Vascular resection	No	0.74 (0.69–0.77)	
Callegaro, Multi-institutional (2021) [29]	Primary Localized Resected	2002–2017	1309	5-year OS	Age Landmark time Grade Resection Occurrence of LR/DR	Yes	0.75–0.85	Dynamic nomogram for longitudinal prognostication 4 centers in 4 countries Digital version available in the Sarculator app (www.sarculator.com)
				5-year DFS	Landmark time Histology Size Grade Multifocality Interaction between all but histology	Yes	0.64–0.72	
Raut, TARPSWG (2019) [33]	Recurrent Resected No metastatic	2002–2011	602	6-year OS	Multifocality Grade Quality of 2nd surgery Histology Age Radiotherapy * Number organs resected *	No	0.7	22 centers in 8 countries * After first surgery
				6-year DFS	Multifocality Grade Quality of 2nd surgery Histology Chemotherapy * Radiotherapy * Number organs resected *	No	0.67	
Zhuang, SHZH (2022) [31]	Primary Localized Resected Liposarcoma	2009–2021	211	1, 2, 5-year OS	Symptoms Needle biopsy Histology LOS	No	0.702	Asian population
				1, 2, 5-year PFS	ASA Score Histology CD classification	No	0.757	
Yiding Li, SEER database (2022) [32]	Primary Localized Resected Liposarcoma	2004–2015	1392	1, 3, 5-year OS	Age Grade Classification SEER stage Surgery	Yes	0.754–0.863	Public database
				1, 3, 5-year CSS	Age Classification SEER Stage AJCC Stage Surgery Tumor Size		0.753–0.829	

SHZH, South Hospital at Zhongshan Hospital, Shanghai; TARPSWG, TransAtlantic RetroPeritoneal Sarcoma Working Group; INT, Istituto Nazionale dei Tumori, Milan; UCLA, University California Los Angeles; MDACC, MD Anderson Cancer Center; MSKCC, Memorial Sloan Kettering Cancer Center; OS, Overall Survival; PFS, Progression Free Survival; DFS, Disease Free Survival; LOS, Length of Stay; CD, Clavien-Dindo; ASA, American Association Anesthesiology; LR, local recurrence; DR, Distant Recurrence; DSD, Disease Specific Death; CSS, Cancer-Specific Survival; SEER, Surveillance, Epidemiology, and End Results. * means that the radiotherapy and the number of resected organs are referred to after the first surgery.

**Table 4 cancers-15-03194-t004:** Follow up according to histology and grade [98,99,100].

Subtype	Follow Up	LR%	DR%	CT Torax	X-ray	CT/MR Abdomen
WDLPS	Every 4–6 m for 5 y, then annualy	60	8	no	yes	yes
DDLPS Grade I–II	Every 4–6 m for 5 y, then annualy	62	28	no	yes	yes
DDLPS Grade III	Every 3–4 m 2–3 y, every 6 m × 2–3 y, then annualy	26	58	yes	no	yes
Pleomorphic LPS	Every 3–4 m 2–3 y, every 6 m × 2–3 y, then annualy	30	50	yes	no	yes
LMS	Every 3–4 m 2–3 y, every 6 m × 2–3 y, then annualy	24	56	yes	no	yes
MPNST	Every 3–4 m 2–3 y, every 6 m × 2–3 y, then annualy	35	15	yes	no	yes
SFT	Every 3–4 m 2–3 y, every 6 m × 2–3 y, then annualy	8	40	yes	no	yes
Other high grade	Every 3–4 m 2–3 y, every 6 m × 2–3 y, then annualy	45	25	yes	no	yes

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
