# Peer review of "Updated Review and Clinical Recommendations for the Diagnosis and Treatment of Patients with Retroperitoneal Sarcoma by the Spanish Sarcoma Research Group (GEIS)"

_cancers, 2023, doi:10.3390/cancers15123194_

Round 1
Reviewer 1 Report
Dear editor,
Thank you for giving me an opportunity to review this manuscript.
This review is about retroperitoneal tumors. This report seems very informative suitable for publication in Cancers because this helps not only general clinicians but also specialized oncologists.
My review is that current version of this review already reached to acceptable levels for Cancers.
I pointed out small points. I would be grateful if this may improve quality of this paper.
1. We sometimes encounter Malignant lymphoma or Ewing sarcoma in retroperitoneal space. At Page 4-5, the authors can add the information about Malignant lymphoma or Ewing sarcoma. Treatment is totally different from spindle cell sarcoma, hence diagnosis is important. Biopsy is completely essential before surgery.
2. At page 6, the authors mention Protein energetic malnutrition. I totally agree with this idea. I would like to ask the authors to state a little more information about the relationship between malnutrition and survival. For example, how risky are patients with malnutrition? Is survival rate significantly different between the patient with good nutrition and those with not?
Author Response
Reviewer #1
My review is that the current version of this review already reached acceptable levels for Cancers. I pointed out small points. I would be grateful if this may improve the quality of this paper.
- We sometimes encounter Malignant lymphoma or Ewing sarcoma in retroperitoneal space. At Pages 4-5, the authors can add information about Malignant lymphoma or Ewing sarcoma. Treatment is totally different from spindle cell sarcoma, hence diagnosis is important. The biopsy is completely essential before surgery.
We agree. We have added a suggested change at the bottom of page 4 (in the biopsy section).
- On page 6, the authors mention Protein energetic malnutrition. I totally agree with this idea. I would like to ask for a little more information about the relationship between malnutrition and survival. For example, how risky are patients with malnutrition? Is the survival rate significantly different between the patient with good nutrition and those with not?
We agree. We have added a suggested change in the Preoperative functional assessment on page 8.

Author Response
Reviewer #2:
- The abbreviation of “differentiated liposarcoma” is not standardized: DDLS in the Introduction and DDLPS in other sections.
You are right, thank you very much for the warning, we have corrected it.
- Regarding postoperative surgery, the intervals between examinations are stated, but the period of follow-up is simply described as that “long-term follow-up of patients beyond 5-10 years is recommended for some high-risk patients and aggressive subtype (Ⅳ, B)”. It would be easier to understand if more specific information were provided.
We agree. We do not have data at present that discriminate those patients in whom long-term follow-up should be performed. For this reason, we have removed "for some high-risk patients and aggressive subtypes".
- Preferably, the differences between this guideline and the conventional guideline should be briefly explained.
We understand that you are referring to the other two important guidelines for retroperitoneal sarcoma (ESMO and TARPSWG).
As for the ESMO guidelines, they address the general management of soft tissue sarcomas without focusing too much on retroperitoneal sarcomas. On the other hand, the TARPSWG has several solid guidelines for multiple key clinical situations: metastatic, recurrent, or primary retroperitoneal sarcomas.
I would like to highlight that in our guideline proposal, we have compiled and summarized key clinical situations providing clear, concise, and practical guidelines/recommendations. Which we consider useful in an integrated multidisciplinary approach to management.
We try to comment briefly on this in the last paragraph of the introduction

Reviewer 3 Report
see file

Author Response
Reviewer #3:
Major comments:
- Last paragraph of the introduction. Is the low prevalence of RPS really the motivation for these guidelines? The reasons given at the end of that paragraph appear more plausible. Please rephrase less ambivalent.
That's a good suggestion because it makes it easier to understand the real purpose of the guide. We modified it. Thank you.
- Methodology section. This comes across as very condensed and leaves many questions unanswered. Which databases were searched for literature? What was the search algorithm? Was it different for each of the questions addressed? Which criteria were used to include or exclude an article in the body of evidence?
We systematically searched data from PUBMED, EMBASE, CENTRAL, OpenGrey, ClinicalTrials.gov, and ProQuest. In each section, we performed different searches, in order to address different questions, prioritizing data with the best evidence level. (We reflect this in the text in more detail)
- Imaging: Is i.v. contrast optional? The word “preferable” in my opinion unnecessarily weakens this recommendation. Obviously, not all patients are able to receive i.v. contrast, but in patients without renal insufficiency or allergy, i.v. contrast is more than a preference.
We agree with your suggestion. We change the word preferable to "highly recommended" and explain in the text why.
- Pathology: Have you considered the EORTC-STBSG recommendations for response evaluation following neoadjuvant therapy? (Wardelmann, E., et al., Evaluation of response after neoadjuvant treatment in soft tissue sarcomas; the European Organization for Research and Treatment of Cancer-Soft Tissue and Bone Sarcoma Group (EORTC-STBSG) recommendations for pathological examination and reporting. Eur J Cancer, 2016. 53: p. 84-95.)
In this guide, we have wanted to focus on the diagnosis of retroperitoneal tumors and we have not included the histological evaluation for sarcomas treated with neoadjuvant therapy, but it is a good idea to mention it.
- Page 10, paragraph 2: This is not “incidence” but risk, because it refers to the individual patient, not the population as a whole.
We agree. We have changed “incidence” to “risk”
Page 10, paragraph 3: What is the evidence for removing the pancreas and spleen in left-sided LPS? If the tumor is sitting caudal in the retroperitoneal space, this is usually not necessary. Even if it is close to the pancreatic tail, the decision may be to preserve the pancreas for reasons of morbidity. In large centers, the pancreatic resection rate for left-sided tumors is around 40%. I doubt they get it 60% wrong.
We agree. We have modified the text.
- Radiotherapy, first paragraph. Is it despite or precisely because of the retrospective nature of studies that radiotherapy was associated with a survival benefit?
We modified it, the objective was to clarify the retrospective nature of the data.
When discussing STRASS, STREXIT (Callegaro D, et al. Preoperative Radiotherapy in Patients With Primary Retroperitoneal Sarcoma: EORTC-62092 Trial (STRASS) Versus Off-trial (STREXIT) Results. Ann Surg. 2022 Jul 14. doi: 10.1097/SLA.0000000000005492) data should be taken into consideration. This study strongly supports the STRASS post hoc analyses and it’s the best we have.
Very good suggestion we have briefly summarized the results of the study in the text.
- It is true that studies in extremity sarcoma cannot be directly extrapolated to RPS. Nonetheless, the significant results of the Italian (ISG), Spanish (GEIS), French (FSG), and Polish (PSG) Sarcoma Groups trial (Gronchi A et al. J Clin Oncol. 2020 Jul 1;38(19):2178-2186. doi: 10.1200/JCO.19.) gives some credibility to the concept of treating high-risk sarcomas with neoadjuvant systemic therapy. The authors may not want to recommend it for RPS, but they should at least discuss this trial briefly.
We have added a minimal commentary on this important study
- Regional hyperthermia has been established in a very limited number of expert centers. Nonetheless, its application is based on an EORTC trial with positive results for DFS, LPFS, and OS (Issels RD et al. Effect of Neoadjuvant Chemotherapy Plus Regional Hyperthermia on Long-term Outcomes Among Patients With Localized High-Risk Soft Tissue Sarcoma: The EORTC 62961-ESHO 95 Randomized Clinical Trial. JAMA Oncol. 2018 Apr 1;4(4):483-492) For OS this was also significant 2 in the RPS subgroup. This RCT cannot be completely ignored. It remains the only RCT with positive results in RPS.
It is briefly described on page 17 (second paragraph). Do you think we should develop it further?
- I am surprised by the lengthy discussion of retrospective studies on RT for irresectable RPS in light of the authors brief dismissal of RT for primary RPS. The body of evidence for RT in irresectable RPS is much weaker.
You are right, all the studies are retrospective. In the absence of studies of better quality, that highlight above the rest, and that the objective of the treatment is palliative with no other option for the patients we have limited ourselves to describe what is there.
Are the authors aware of any clinical trials investigating HIPEC in RPS? Will such a trial ever be feasible?
If we add it to the text. Good suggestion that improves the guideline!
Minor comments:
Reference center- heading: there is a punctuation mark or colon missing. Corresponding recommendation: it should be “network” without an “s”
There is an arrow flying around at the bottom of page 6.
Page 10, first line: there is a space missing after “centers”
What is “maximum stress surgery”? I do not know this term.
Advanced disease: what is “PRS”? typo?
Follow-up: The word “be” can be omitted.
You are right, thank you very much for the warning, we have corrected it.
